# Accented Speech Recognition With Accent-specific Codebooks

**Darshan Prabhu**[‡], **Preethi Jyothi**[‡], **Sriram Ganapathy**[§], **Vinit Unni**[‡]

[‡]Indian Institute of Technology Bombay, Mumbai, India
[§]Indian Institute of Science, Bangalore, India

{darshanp,pjyothi,vinit}@cse.iitb.ac.in[‡], sriramg@iisc.ac.in[§]

## Abstract

Speech accents pose a significant challenge to state-of-the-art automatic speech recognition (ASR) systems. Degradation in performance across underrepresented accents is a severe deterrent to the inclusive adoption of ASR. In this work, we propose a novel accent adaptation approach for end-to-end ASR systems using cross-attention with a trainable set of codebooks. These learnable codebooks capture accent-specific information and are integrated within the ASR encoder layers. The model is trained on accented English speech, while the test data also contained accents which were not seen during training. On the Mozilla Common Voice multi-accented dataset, we show that our proposed approach yields significant performance gains not only on the seen English accents (up to 37% relative improvement in word error rate) but also on the unseen accents (up to 5% relative improvement in WER). Further, we illustrate benefits for a zero-shot transfer setup on the L2Artic dataset. We also compare the performance with other approaches based on accent adversarial training.

## 1 Introduction

Accents in speech typically refer to the distinctive way in which the words are pronounced by diverse speakers. While a speaker's accent may be primarily derived from their native language, speech accents are also influenced by various other factors related to the geographic location, educational background, socio-economic and socio-linguistic factors like race, gender and cultural diversity (Benzeghiba et al., 2007). It is therefore infeasible to build automatic speech recognition (ASR) systems which comprehensively cover speech accents during training. In such scenarios, novel speech accents continue to have an adverse effect on ASR performance (Beringer et al., 1998; Aksënova et al., 2022). While humans effectively recognize speech from new and unseen accents (Clarke and Garrett,

2004), ASR systems show substantial degradation in performance when dealing with new accents that are unseen during training (Chu et al., 2021).

Prior works attempting to address accent-related challenges for ASR can be categorized into three groups: i) multi-accent training (Huang et al., 2014; Elfeky et al., 2016), ii) accent-aware training using accent embeddings (Jain et al., 2018) or adversarial learning (Sun et al., 2018), and iii) accent adaptation using supervised (Rao and Sak, 2017; Winata et al., 2020) or unsupervised techniques (Turan et al., 2020). While partial success has been achieved using most of these approaches, the development of robust speech recognition systems that are invariant to accent differences in training and test remains a challenging problem.

In this work, we propose a new codebook based technique for accent adaptation of state-of-the-art Conformer-based end-to-end (E2E) ASR models (Gulati et al., 2020). For each of the accents observed in the training data, we define a codebook with a predefined number of randomly-initialized vectors. These accent codes are integrated with the self-attended representations in each encoder layer via the cross-attention mechanism, similar to the perceiver framework (Jaegle et al., 2021). The ASR model is trained on multi-accented data with standard end-to-end (E2E) ASR objectives. The codes capture accent-specific information as the training progresses. During inference, we propose a beam-search decoding algorithm that searches over a combined set of hypotheses obtained by using each set of accent-specific codes (once for each seen accent) with the trained ASR model. On the Mozilla Common Voice (MCV) corpus, we observe significant improvements on both seen and new accents at test-time compared to the baseline and existing supervised accent-adaptation techniques.

Our main contributions are:

- We propose a new accent adaptation technique for Conformer-based end-to-end ASR models

using cross-attention over a set of learnable codebooks. Our technique comprises learning accent-specific codes during training and a new beam-search decoding algorithm to perform an optimized combination of the codes from the seen accents. We demonstrate significant performance improvements on both seen and unseen accents over competitive baselines on the MCV dataset.

- Even on a zero-shot setting involving a new accented evaluation set, L2-Arctic (Zhao et al., 2018), we show significant improvements using our codebooks trained using MCV.

- We publicly release our train/development/test splits spanning different seen and unseen accents in the MCV corpus. Reproducible splits on MCV have been entirely missing in prior work and we hope this will facilitate fair comparisons across existing and new accent-adaptation techniques.[1]

## 2 Related Work

Traditional cascaded ASR systems (García-Moral et al., 2007) handled accents by either modifying the pronunciation dictionary (Humphries and Woodland, 1997; Weninger et al., 2019) or modifying the acoustic model (Fraga-Silva et al., 2014; Yoo et al., 2019). More recent work on accented ASR has focused on building end-to-end accent-robust ASR models. Towards this, there are two sets of prior works: *Accent-agnostic* approaches and *Accent-aware* approaches.

**Accent-agnostic ASR.** Such approaches force the model to disregard the accent information present in the speech and focus only on the underlying content. Prior work based on this approach uses adversarial training (Ganin et al., 2015) or similarity losses. Using domain adversarial training, with the discriminator being an accent classifier, has shown significant improvements over standard ASR models (Sun et al., 2018). Pre-training the accent classifier (Das et al., 2021b) and clustering-based accent relabelling (Hu et al., 2020) have also led to further performance improvements. The use of generative adversarial networks for this task has also been explored (Chen et al., 2019). Rather than being explicitly domain adversarial, other accent

agnostic approaches use cosine losses (Unni et al., 2020) or contrastive losses (Khosla et al., 2020; Han et al., 2021) to make the model accent neutral. These losses force the model to output similar representations for inputs with the same underlying transcript.

**Accent-aware ASR.** Accent-aware approaches feed the model additional information about the accent of the input speech. Early work in this category focused on using the multi-task learning (MTL) paradigm (Zheng et al., 2015; Jain et al., 2018; Das et al., 2021a) that jointly trains accent-specific auxiliary tasks with ASR. Different types of embeddings like i-vectors (Saon et al., 2013; Chen et al., 2015), dialect symbols (Li et al., 2017), embeddings extracted from TDNN models (Jain et al., 2018) or from wav2vec2 models trained as classifier (Li et al., 2021a; Deng et al., 2021) have also been explored for accented ASR. Many simple ways of fusing accent information with the input speech have been previously investigated. This fusion can either be a sum (Jain et al., 2018; Viglino et al., 2019; Li et al., 2021a), a weighted sum (Deng et al., 2021) or a concatenation (Li et al., 2021a,b). Few works also explore the possibility of merging both accent-aware and accent-agnostic techniques within the same model (Zhou et al., 2023). Our work also proposes an accent-aware approach. However, unlike prior work that focuses on prefetched accent information, we learn accent information embedded within codebooks during training. Additionally, instead of simply concatenating input speech with accent embeddings, we propose a learned fusion of accent information with speech representations using cross-attention. Prior work by Deng et al. (2021) demonstrates fine-grained integration of accent information. However, our proposed framework integrates accent information as part of end-to-end training resulting in robust adaptation.

## 3 Methodology

**Base model.** Our base architecture uses the standard joint CTC-Attention framework (Kim et al., 2016) with an encoder (ENC), a decoder (DEC-ATT), and a Connectionist Temporal Classification (CTC) (Graves et al., 2006) module (DEC-CTC). For a given speech input $\mathbf{x} = \{x_1, \ldots, x_T\}$, the encoder ENC generates contextualized representations $\mathbf{h} = \text{ENC}(\mathbf{x}) = \{h_1, \ldots, h_T\}$. The encoder representations $\mathbf{h}$ are further used by DEC-ATT

---

[1]The MCV data splits and codebase are available at: https://github.com/csalt-research/accented-codebooks-asr.

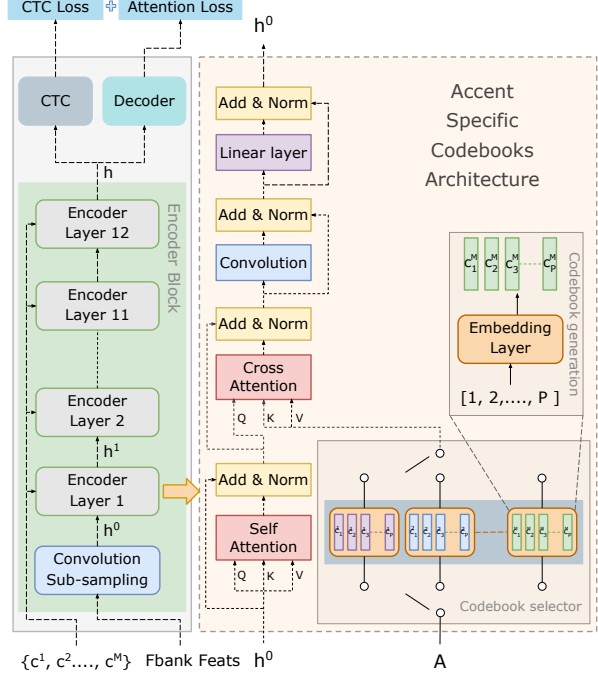

Figure 1: Overview of our proposed architecture integrating accent codebooks into encoder layers via cross-attention. $A$ represents the accent label for a training instance and $\{c^1, c^2, \ldots c^M\}$ is the collection of accent-specific codebooks .

and DEC-CTC to jointly predict the output token sequence $\mathbf{y} = \{y_1, \ldots, y_j, \ldots, y_U\}$. DEC-ATT is an autoregressive decoder that maximizes the conditional likelihood of producing an output token $y_j$ given $\mathbf{h}$ and the previous labels $y_1, \ldots, y_{j-1}$. In contrast, DEC-CTC uses CTC to maximize the likelihood of $\mathbf{y}$ given $\mathbf{h}$ by marginalizing over all alignments. The encoder is implemented using Conformer layers (Gulati et al., 2020) and the decoder is implemented using Transformer layers (Vaswani et al., 2017).

For our proposed technique, we introduce the following three essential modifications to the base architecture: i) Constructing codebooks that can encode accent-specific information (Section 3.1). ii) Enabling fine-grained integration of accent information with a Conformer-based ASR model using cross-attention (Section 3.2). iii) Modifying beam search decoding for inference in the absence of accent labels at test-time (Section 3.3).

## 3.1 Codebook Construction

Consider $M$ seen accents, which are observed during training. We generate $M$ codebooks, one per accent, where the $i^{\text{th}}$ codebook learns latent codes specific to the $i^{\text{th}}$ accent. During training, we use a deterministic gating scheme to select the codebook specific to the underlying accent of the training example. To support the selection of a single accent codebook during inference, when the accent labels for the test utterances are unknown, we modify the beam-search decoder to search across all seen accents. We found such a hard gating to be critical to achieve ASR performance improvements. In Section 6, we compare the proposed model with a soft gating mechanism that works with a standard beam-search decoding.

Each codebook contains $P$ $d$-dimensional vectors that we refer to as codebook entries. The entries belonging to the $i^{\text{th}}$ codebook are generated as:

$$\mathbf{c}^i = \{c_1^i, \ldots, c_P^i\} = \texttt{Embedding}([1, 2, \ldots, P])$$

where Embedding is a standard embedding layer. In the following sections, we use $\mathbf{c} = \{c_1, \ldots, c_P\}$ to refer to the codebook corresponding to the underlying accent label for a given training example.

## 3.2 Encoder with Accent Codebooks

Figure 1 illustrates the overall architecture with the proposed integration of a codebook into each encoder layer via a cross-attention sub-layer. We will refer to this new accent-aware encoder module as $\text{ENC}_a = \{\text{ENC}_a^1, \ldots, \text{ENC}_a^L\}$ that consists of a stack of $L$ identical Conformer layers. The $i^{\text{th}}$ encoder layer $\text{ENC}_a^i$ takes both $\mathbf{h}^{i-1}$ and $\mathbf{c}$ as inputs, and produces $\mathbf{h}^i$ as output. Codebook $\mathbf{c}$ is shared across all the encoder layers. All the vectors involved in the computation of attention scores are $d = 256$ dimensional.

A cross-attention sub-layer integrates accent information from codebook $\mathbf{c}$ into each encoder layer. This sub-layer takes both self-attended contextualized representations $\mathbf{H}$ and the codebook $\mathbf{c}$ as its inputs and generates codebook-specific information relevant to the speech frames of this contextual representation. More formally, the operations within encoder layer $\text{ENC}_a^i$ can be written as follows:

$$\hat{\mathbf{H}} = \text{MultiHeadAttn}_{\text{self}}(\mathbf{h}^{i-1}, \mathbf{h}^{i-1}, \mathbf{h}^{i-1})$$
$$\mathbf{H} = \text{NormLayer}_{\text{self}}(\mathbf{h}^{i-1} + \hat{\mathbf{H}})$$
$$\hat{\mathbf{C}} = \text{MultiHeadAttn}_{\text{cb}}(\mathbf{H}, \mathbf{c}, \mathbf{c})$$
$$\mathbf{C} = \text{NormLayer}_{\text{cb}}(\mathbf{H} + \hat{\mathbf{C}})$$
$$\hat{\mathbf{J}} = \text{Convolution}(\mathbf{C})$$
$$\mathbf{J} = \text{NormLayer}_{\text{conv}}(\mathbf{C} + \hat{\mathbf{J}})$$
$$\hat{\mathbf{h}}^i = \text{Linear}_{\text{pw}}(\mathbf{J})$$
$$\mathbf{h}^i = \text{NormLayer}_{\text{linear}}(\mathbf{J} + \hat{\mathbf{h}}^i)$$

where the equations colored in purple highlight our changes to the standard Conformer encoder layer. The above equations can be viewed as a stack of four independent blocks, each having a residual connection and being separated by layer normalization.

$\text{MultiHeadAttn}(Q, K, V)$ refers to a standard multi-head attention module (Vaswani et al., 2017) with $Q$, $K$ and $V$ denoting queries, keys and values respectively. $\text{MultiHeadAttn}_{\text{self}}$ is a self-attention module where each frame of $\mathbf{h}^{i-1}$ attends to every other frame, thus adding contextual information. Convolution is a stack of three convolution layers: a depth-wise convolution sandwiched between two point-wise convolutions, each having a single stride. The input and output of the Convolution block are $d$-dimensional vectors. A position-wise feed-forward layer $\text{Linear}_{\text{pw}}$ is made up of two linear transformations with a ReLU activation. This takes a $d$-dimensional output from the convolution module as input and produces a $d$-dimensional output vector with a hidden layer of 2048 dimensions.

$\text{MultiHeadAttn}_{\text{cb}}(\mathbf{H}, \mathbf{c}, \mathbf{c})$ is our proposed cross-attention module over codebook entries, where each frame of input $\mathbf{H}$ attends to all the entries in the codebook $\mathbf{c}$ to generate attention scores. These attention scores are further used to generate frame-relevant information $\hat{\mathbf{C}}$ as a weighted average of codebook entries. We elaborate further on the attention computation for a single attention head in $\hat{\mathbf{C}}$.[2] Let $\mathbf{H}_j$ refer to the $j^{\text{th}}$ frame in $\mathbf{H}$. The attention distribution $\{\alpha_{j,1}, \ldots \alpha_{j,P}\}$, where $\alpha_{j,k}$ is the attention probability given by $\mathbf{H}_j$ to the $k^{\text{th}}$ codebook entry, is computed as:

$$\{\alpha_{j,1}, \alpha_{j,2}, \ldots, \alpha_{j,P}\} =$$
$$\text{softmax}\left(\frac{(W_q^i \mathbf{H}_j)(W_k^i \mathbf{c})^T}{\sqrt{d}}\right)(W_v^i \cdot \mathbf{c})$$

where $W_q^i$, $W_k^i$ and $W_v^i \in \mathbb{R}^{d \times d}$ are learned projection matrices. These attention scores are further used to generate the weighted average of codebook entries in $\hat{\mathbf{C}}$:

$$\hat{\mathbf{C}}_j = \sum_{k=1}^{P} \alpha_{j,k} \cdot \mathbf{c}_k$$

The cross-attention sublayer is further modified with a residual connection and layer normalization to generate the final codebook-infused representations in $\mathbf{C}$.

---

[2]In our experiments, self-attention uses four attention heads for the encoder and cross-attention with the codebook uses a single attention head.

**Algorithm 1:** Inference algorithm that performs joint beam-search over all accents. Our modifications to the standard beam search (Meister et al., 2020) are highlighted . Each beam entry is a triplet $\langle s, y, A \rangle$ where $A$ refers to a seen accent. $\text{score}_A()$ is a modified scoring function which uses the codebook for accent $A$ during the forward pass.

> **Input :** $\mathbf{x}$: speech input
> $\mathcal{V}$: list of vocabulary tokens
> $n_{\max}$: maximum hypothesis length
> $k$: maximum beam width
> $y$: output prediction so far
> $\text{score}_A(.,.,.)$: scoring function
>
> 1   $\mathcal{B}_0 = \{\langle 0, \texttt{<sos>}, 1\rangle \ldots, \langle 0, \texttt{<sos>}, M\rangle\}$
> 2   **for** $t \in \{1, \ldots, n_{\max} - 1\}$ **do**
> 3     $\mathcal{B} \leftarrow \phi$
> 4     **for** $\langle s, y, A \rangle \in \mathcal{B}_{t-1}$ **do**
> 5       **if** $y.\text{last}() == \texttt{<eos>}$ **then**
> 6         $\mathcal{B}.\text{add}(\langle s, y, A \rangle)$
> 7         continue
> 8       **for** $v \in \mathcal{V}$ **do**
> 9         $s \leftarrow \text{score}_A(\mathbf{x}, y \circ v, A)$
> 10        $\mathcal{B}.\text{add}(\langle s, y \circ v, A \rangle)$
> 11     $\mathcal{B}_t = \mathcal{B}.\text{top}(k)$
> 12   **return** $\mathcal{B}$

### 3.3 Modified Beam-Search Algorithm

Since we do not have access to accent labels at test-time, we rely on either using a classifier to predict the accent or modifying beam-search to accommodate the prediction streams generated by all seen accent choices. Due to a large imbalance in the accent distribution during training with certain seen accents dominating the training set, we find the classifier to be ineffective during inference. We elaborate on this further in Section 6.

Figure 1 shows our inference algorithm that performs a joint beam search over all the seen accents. Each beam entry is a triplet that expands each hypothesis using each seen accent. Scores for each seen accent are computed using a forward pass through our ASR model by invoking the codebook specific to the accent. The beam width threshold $k$ is then applied to expanded predictions across all seen accents.

# 4 Experimental Setup

## 4.1 Datasets

All our experiments are conducted on the MCV_ACCENT dataset extracted from the "validated" split of the Mozilla Common Voice English (en) corpus (Ardila et al., 2019). Overall, 14 English accents present in MCV_ACCENT are divided into two groups of *seen* and *unseen* accents. Table 1 lists the accents belonging to each of these groups.

| Seen Accents | Unseen Accents | | | |
|---|---|---|---|---|
| Australia (AUS) | Africa | (AFR) | Malaysia | (MAL) |
| Canada (CAN) | New Zealand | (NWZ) | Hong Kong | (HKG) |
| England (GBR) | Philippines | (PHL) | India | (IND) |
| Scotland (SCT) | Singapore | (SGP) | Ireland | (IRL) |
| US (USA) | Wales | (WLS) | | |

Table 1: List of 5 seen and 9 unseen accents in MCV_ACCENT corpus.

We create train, dev, and test splits that are speaker-disjoint. We construct two train sets, MCV_ACCENT-100 and MCV_ACCENT-600, comprising approximately 100 hours and 620 hours of labeled accented speech, respectively. Since MCV_ACCENT consists of many utterances that correspond to the same underlying text prompts, a careful division into dev/test sets that are disjoint in transcripts from the train set was performed. We train and validate the ASR models only on the seen accents, while the test data consists of both seen and unseen accents. The detailed statistics of our datasets are given in Table 2. For a quick turnaround, most of our experiments were conducted on MCV_ACCENT-100. For experiments in Section 5.3, we use the 620-hour MCV_ACCENT-600. All the data splits mentioned above are available in our codebase, which should enable direct comparisons across accent adaptation techniques. More details about the construction of the datasets are provided in Appendix A.

## 4.2 Models and Implementation Details

We use the ESPnet toolkit (Watanabe et al., 2018) for all our ASR experiments. As is standard practice, we further add 3-way speed perturbation to our dataset before training. We use the default configurations specified in the `train_conformer.yaml` file provided in the ESPnet toolkit[3] to train a Conformer model comprising 12 encoder layers

[3]https://tinyurl.com/2wexthds

| Accent | Train (in hours) | | Dev (in hours) | Test (in hours) |
|---|---|---|---|---|
| | MA-100 | MA-600 | | |
| Australia | 6.95 | 45.36 | 4.33 | 0.46 |
| Canada | 6.79 | 41.13 | 1.16 | 1.21 |
| England | 19.51 | 119.9 | 3.22 | 1.65 |
| Scotland | 2.69 | 16.21 | 0.23 | 0.16 |
| US | 64.12 | 400.1 | 8.32 | 4.87 |
| Africa | – | – | | 1.71 |
| Hongkong | – | – | | 0.52 |
| India | – | – | | 0.58 |
| Ireland | – | – | | 1.94 |
| Malaysia | – | – | | 0.39 |
| Newzealand | – | – | | 2.11 |
| Philippines | – | – | | 0.90 |
| Singapore | – | – | | 0.64 |
| Wales | – | – | | 0.27 |

Table 2: Data splits for experiments. MA-100 and MA-600 refers to datasets MCV_ACCENT-100 and MCV_ACCENT-600 datasets respectively.

and 6 decoder layers using joint CTC-Attention loss (Kim et al., 2016). We use four attention heads to attend over 256-dimensional tensors. The position-wise linear layer operates with 2048 dimensions. We train the model for 50 epochs using 80-dimensional filter-bank features with pitch. In all our experiments, we apply a stochastic depth rate of 0.3 which we found to yield an absolute 2% WER improvement, compared to a baseline system without this regularization enabled. During inference, we use a two-layer RNN language model (Mikolov et al., 2010) trained for 20 epochs with a batch size of 64. We conducted all our experiments on NVIDIA RTX A6000 GPUs.

# 5 Experiments and Results

Table 3 shows word error rates (WERs) comparing our best system (codebook attend (CA) system) with five approaches: 1. Transformer baseline (Dong et al., 2018). 2. Conformer baseline (Gulati et al., 2020). 3. Adding i-vector features (Chen et al., 2015)[4] to the filterbank features, as input to the Conformer baseline. 4. Conformer jointly trained with an accent classifier using multi-task learning (Jicheng et al., 2021). 5. Conformer with Domain Adversarial Training (DAT) (Das et al., 2021b), with an accent classifier at the 10th encoder layer.

[4]Every frame in i-vectors represents a second. Since the input filterbank features span 25msec, we repeat the same i-vector frame for four consecutive feature frames. Adding, as opposed to concatenating, the i-vector frames was found to perform better.

| Method | Aggregated | | | Seen Accents | | | | | Unseen Accents | | | | | | | | |
|---|---|---|---|---|---|---|---|---|---|---|---|---|---|---|---|---|---|
| | All | Seen | Unseen | AUS | CAN | UK | SCT | US | AFR | HKG | IND | IRL | MAL | NWZ | PHL | SGP | WLS |
| Trans. (Dong et al., 2018) | 22.7 | 17.3 | 28.0 | 18.1 | 17.8 | 19.7 | 18.5 | 16.3 | 25.9 | 32.0 | 35.4 | 25.3 | 36.2 | 23.8 | 31.5 | 38.8 | 21.0 |
| Conf. (Gulati et al., 2020) | 18.9 | 14.0 | 23.7 | 13.8 | **15.0** | 15.7 | **13.4** | 13.3 | 21.5 | 27.2 | 29.4 | 21.4 | 32.2 | 19.9 | 26.1 | 34.7 | 17.9 |
| I-vector (Chen et al., 2015) | 18.9 | 14.1 | 23.6 | 13.9 | 15.0 | 16.1 | 14.6 | 13.3 | 21.7 | 27.2 | 29.5 | 21.2 | **31.7** | 19.3 | 27.2 | **33.8** | 18.0 |
| MTL (Jicheng et al., 2021) | 18.9 | 14.1 | 23.7 | 14.7 | 15.1 | 16.1 | 13.7 | 13.2 | 21.8 | 28.1 | **29.1** | 21.5 | 33.0 | 19.4 | 26.5 | 34.2 | 18.1 |
| DAT (Das et al., 2021b) | **18.7** | **14.0** | **23.4** | 13.3 | 15.3 | **15.7** | 15.5 | **13.1** | **21.1** | **27.0** | 29.5 | **21.1** | 32.2 | **19.2** | **26.0** | 34.4 | **17.9** |
| CA | 18.2† | 13.6 | 22.9 | 11.5 | 14.8 | 14.9 | 9.7 | 13.1 | 21.0 | 25.7 | 29.1 | 20.7 | 30.9 | 18.5 | 25.8 | 33.7 | 17.9 |

Table 3: Comparison of the performance (WER % ) of our architecture (codebook attend (CA)) with baseline and other techniques on the MCV-ACCENT-100 dataset. Numbers in bold denote the best across baselines, and the green highlighting ▢ denotes the best WER across all experiments. Ties are broken using overall WER. CA: Codebook attend - cross-attention applied at all layers with 50 entries in each learnable codebook. † indicates statistically significant results compared to DAT (at $p$ <0.001 using MAPSSWE test (Gillick and Cox, 1989)).

From Table 3, we observe that the Conformer baseline performs significantly better than the Transformer baseline. Adding i-vectors and multi-task training with an auxiliary accent classifier objective perform equally well and are comparable to the Conformer baseline, while using DAT improves over the Conformer baseline. Our system significantly outperforms DAT (at $p < 0.001$ using the MAPSSWE test (Gillick and Cox, 1989)) and achieves the lowest WERs across all the seen and unseen accents. We use 50 codebook entries for each accent and incorporate accent codebooks into each of the 12 encoder layers. Unless specified otherwise, we will use this configuration in all subsequent experiments. Further ablations of these choices will be detailed in Section 5.5.

## 5.1 Zero-shot Transfer

| Method | All | Accents | | | | | |
|---|---|---|---|---|---|---|---|
| | | ARA | HIN | KOR | MAN | SPA | VIA |
| Conformer | 33.3 | 30.4 | 30.4 | 26.9 | 37.9 | 30.3 | 43.5 |
| I-vector | 33.6 | 31.0 | 31.2 | 27.2 | 38.0 | 30.4 | 43.9 |
| MTL | 33.4 | 30.4 | 30.6 | 26.9 | 38.7 | 30.1 | 43.7 |
| DAT | 33.5 | 30.7 | 30.8 | 26.8 | 38.3 | 30.1 | 43.9 |
| CA | 32.6† | 29.5 | 30.4 | 26.2 | 37.1 | 29.3 | 42.8 |

Table 4: Comparison of the zero-shot performance (WER %) of our architecture with other techniques on L2Arctic dataset. † indicates a statistically significant improvement ($p$ <0.001 using MAPSSWE test) using codebook attend (CA) w.r.t. the Conformer baseline.

To further validate the efficacy of our proposed approach using accent-specific codebooks, we perform zero-shot evaluations on the L2Arctic dataset. We note here that we do not use any L2Arctic data for finetuning; our ASR model is trained on MCV_ACCENT-100. Such a zero-shot evaluation helps ascertain whether our codebooks transfer well across datasets. The L2Arctic dataset (Zhao et al., 2018) comprises English utterances spanning six non-native English accents namely Arabic (ARA), Hindi (HIN), Korean (KOR), Mandarin (MAN), Spanish (SPA), and Vietnamese (VIA). Table 4 shows WERs achieved by our system in comparison to the baseline and other techniques. Our proposed method significantly outperforms all these approaches on every single accent ($p < 0.001$ using the MAPSSWE test (Gillick and Cox, 1989)).

## 5.2 Effect of Training Data Size

| Method | Overall | Seen | Unseen |
|---|---|---|---|
| Conf. (Gulati et al., 2020) | 9.75 | **6.04** | 13.46 |
| I-vector (Chen et al., 2015) | 10.05 | 6.40 | 13.69 |
| MTL (Jicheng et al., 2021) | 10.02 | 6.33 | 13.70 |
| DAT (Das et al., 2021b) | 9.73 | 6.12 | 13.33 |
| $CA_{L \in (1,...,12)}(P = 50)$ | 9.63 | 6.22 | 13.03 |
| $CA_{L \in (1,...,12)}(P = 200)$ | 9.59 | 6.20 | 12.98 |
| $CA_{L \in (1,...,12)}(P = 500)$ | **9.55** | 6.19 | **12.92** |

Table 5: Comparison of the performance (WER %) of our approach with other methodologies on MCV_ACCENT-600 dataset.

Table 5 compares our proposed system with DAT and Conformer on the 600-hour MCV_ACCENT dataset. Compared to the Conformer baseline and the DAT, the proposed CA approach shows a steady improvement over unseen accents, while resulting in a minor drop in performance on the seen accents.

## 5.3 Effect of Number of Parameters

| Method | # of params | Overall | Seen | Unseen |
|---|---|---|---|---|
| Conf. | 43M | 18.87 | 14.05 | 23.67 |
| Conf. w/ ↑ encoder units | 46M | 18.89 | 14.02 | 23.74 |
| Conf. w/ ↑ attention dim | 46M | 18.77 | 14.02 | 23.51 |
| $CA_{L \in (1,...,12)}(P = 50)$ | 46M | **18.22** | **13.57** | **22.86** |

Table 6: Comparison of the performance (WER %) of our approach with parameter-equivalent variants of the Conformer baseline on MCV_ACCENT-100.

To discount the possibility that improvements

using our proposed model could be attributed to an increase in the number of parameters, in Table 6, we compare our proposed system with multiple variants of the baseline Conformer model (referred to as Conf. in Table 3) where parameters are increased to be commensurate with our proposed model by either (1) Increasing the number of encoder units (from $2048 \rightarrow 2320$) or (2) Increasing the dimension used for attention computation (from $256 \rightarrow 272$). We observe a slight improvement over the standard baseline when the attention dimension is increased. However, compared to all these baselines, our proposed model still shows a statistically significant improvement at $p < 0.001$.

## 5.4 Balanced versus Imbalanced Dataset

| Method | Overall | Seen | Unseen |
|---|---|---|---|
| Conformer | 19.30 | 14.73 | 23.86 |
| $\text{CA}_{L \in (1,...,12)}(P = 50)$ | **18.88** | **14.61** | **23.13** |

Table 7: Comparison of the performance (WER %) of our approach with Conformer baseline on an accent balanced MCV_ACCENT-100 dataset.

To check the effectiveness of our approach on a balanced dataset, in Table 7, we compare our proposed system with the Conformer baseline on a 100-hour accent-balanced data split. Even on such a balanced dataset, our architecture shows a statistically significant improvement (at $p=0.005$) compared to the baseline.

## 5.5 Ablation Studies

We present two ablation analyses examining the effect of changing the number of accent-specific codebook entries ($P$) and the effect of applying cross-attention at different encoder layers.

The first five rows in Table 8 refer to the addition of codebooks to all encoder layers via cross-attention with varying accent-specific codebook sizes ($P$) ranging from 25 to 500. As $P$ increases, the experiments show improved performance on seen accents but degrades on the unseen accents, indicating that the codebooks begin to overfit to the seen accents. Our best-performing system with $P = 50$ performs well on seen accents while also generalizing to the unseen accents. As expected, using lower-capacity codebooks ($P = 25$) shows performance degradation.

The next five rows in Table 8 refer to codebooks with cross-attention introduced at varying encoder layers. The number of codebook entries is fixed at

| Method | Overall | Seen | Unseen |
|---|---|---|---|
| $\text{CA}_{L \in (1,...,12)}(P = 25)$ | 18.33 | 13.76 | 22.89 |
| $\text{CA}_{L \in (1,...,12)}(P = 50)$ | **18.22** | **13.57** | **22.86** |
| $\text{CA}_{L \in (1,...,12)}(P = 100)$ | 18.36 | 13.85 | 22.86 |
| $\text{CA}_{L \in (1,...,12)}(P = 200)$ | 18.41 | 13.69 | 23.12 |
| $\text{CA}_{L \in (1,...,12)}(P = 500)$ | 18.39 | 13.68 | 23.09 |
| $\text{CA}_{L \in (1,...,4)}(P = 50)$ | **18.30** | 13.95 | **22.64** |
| $\text{CA}_{L \in (1,...,8)}(P = 50)$ | 18.31 | **13.86** | 22.75 |
| $\text{CA}_{L \in (9,...,12)}(P = 50)$ | 18.92 | 14.24 | 23.59 |
| $\text{CA}_{L \in (5,...,12)}(P = 50)$ | 18.45 | 13.84 | 23.05 |
| $\text{CA}_{L \in (1,...,12)}(P_{\text{rand}} = 50)$ | 18.30 | 13.65 | 22.95 |

Table 8: Comparison of the performance (WER %) of different variants of our architecture. $\text{CA}_{L \in (i,...,j)}(P = k)$: Codebook attention applied at all layers from $i$ to $j$ with $k$ entries per accent codebook. $\text{CA}_{L \in (i,...,j)}(P_{\text{rand}} = k)$: Similar to the previous setup, but with codebooks frozen during training. Accent-wise WER is shown in Appendix B and a few select examples are highlighted in Appendix C.

50. Since accent effects can be largely attributed to acoustic differences, we see that the early encoder layers closer to the speech inputs benefit most from the codebooks. Adding codebooks only to the last four or eight encoder layers is not beneficial.

Randomly initialized codebooks were observed to be as useful as learnable codebooks for self-supervised representation learning in Chiu et al. (2022). Motivated by this result, we experiment with randomly-initialized accent-specific codebooks that are not learned during training. The last row of Table 8 shows that random codebooks only cause a slight degradation in performance compared to the best performing system, echoing the observations in Chiu et al. (2022).

## 5.6 Inference with a Single Accent

To understand the effectiveness of accent-specific codebooks, we conduct five experiments by committing to a single seen accent during inference. That is, we decode all the test utterances using a fixed accent label. Table 9 shows results from inferring with a single accent across both seen and unseen accents. For the seen accents, the diagonal contains the lowest WERs indicating that the information learned in our codebooks benefits the accented samples. Furthermore, similar accents, from geographically-close regions, benefit each other. The New Zealand accented English speech achieves the best WERs using Australian accent specific codebooks, Hong Kong, Indian, Philippines and Singapore accented test utterances prefer

| Accent used | Seen Accents | | | | | Unseen Accents | | | | | | | | |
|---|---|---|---|---|---|---|---|---|---|---|---|---|---|---|
| | **AUS** | **CAN** | **UK** | **SCT** | **US** | **AFR** | **HKG** | **IND** | **IRL** | **MAL** | **NWZ** | **PHL** | **SGP** | **WLS** |
| Australia | **11.5** | 19.5 | 17.0 | 18.1 | 17.4 | 22.0 | 29.8 | 32.5 | 24.3 | 33.7 | **18.7** | 30.1 | 37.8 | 21.1 |
| Canada | 20.5 | **14.7** | 20.0 | 15.7 | 13.5 | 24.5 | 27.4 | 29.6 | **21.4** | 32.7 | 25.7 | 26.6 | 35.4 | 21.8 |
| England | 13.8 | 17.7 | **15.0** | 14.4 | 16.2 | **21.5** | 27.0 | 29.9 | 22.0 | **32.3** | 21.1 | 27.1 | 34.8 | **18.0** |
| Scotland | 20.7 | 17.8 | 19.1 | **10.2** | 16.4 | 24.4 | 28.2 | 33.5 | 22.6 | 34.4 | 25.7 | 29.0 | 36.6 | 21.3 |
| US | 20.2 | **14.7** | 19.4 | 15.5 | **13.2** | 23.4 | **27.0** | **28.1** | 21.7 | 32.4 | 24.7 | **25.8** | **34.3** | 22.2 |

Table 9: Comparison of the performances (WER%) of inferences done using fixed accent labels.

US accented codebooks, and Wales accent achieves its best results using England-specific codebooks.

The WER results achieved by our best-performing system in Table 3 are much lower than the best WER results achieved in these single-accent experiments. This indicates that one cannot directly map an unseen accent to an appropriate seen accent and therefore, making this decision independently for each utterance (as we propose to do in the joint beam search) is crucial.

### 5.7 Beam-Search Decoding Variants

All the results reported thus far use a joint beam search decoding. Table 10 shows a comparison of our proposed joint beam search (elaborated in Section 3.3) with other beam-search variants incurring varying inference overheads. $\mathbb{B}_0$ in Table 10

| Method | All | Seen | Unseen | Inference Time |
|---|---|---|---|---|
| $\mathbb{B}_0$: Standard beam search | 18.87 | 14.05 | 23.67 | 1.0 |
| $\mathbb{B}_1$: $M$ full beam searches | **18.10** | **13.48** | **22.71** | 5.02 |
| $\mathbb{B}_2$: $M$ split beam searches | 18.30 | 13.61 | 22.97 | 1.14 |
| $\mathbb{B}_3$: Joint beam search | 18.22 | 13.57 | 22.86 | 1.16 |

Table 10: WER (%) of various inference algorithms described in section 5.7 on MCV_ACCENT-100 setup. Inference time gives a relative comparison of the time taken by each decoding variant with the standard beam search as the reference.

refers to a standard beam-search decoding over the Conformer baseline with a beam width of $k$. The setting $\mathbb{B}_1$ and $\mathbb{B}_2$ refer to running beam-search $M$ times, once for each seen accent and picking the best-scoring hypothesis among all predictions. For $\mathbb{B}_1$ setting, we use a beam width of $k$ for each seen accent. Naturally, this incurs a large decoding overhead with a factor of $M$ increase in inference time and changes the effective beam width to $Mk$. In the $\mathbb{B}_2$ setting, we divide the beam width into $M$ parts, each occupied by a specific accent, thus making the effective beam width $k/M$. The setting $\mathbb{B}_1$ performs the best, but significantly increases inference overhead. The $\mathbb{B}_2$ setting is efficient but under-performs due to all accents being given an

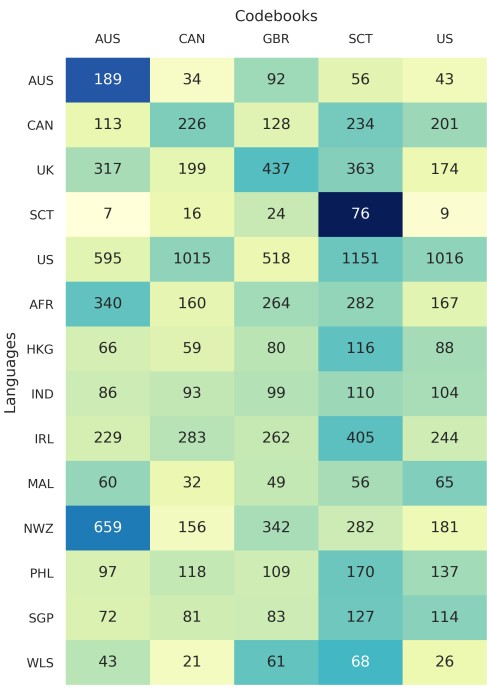

Figure 2: Heatmap showing which codebooks are chosen during inference across seen and unseen accents. For example, the third cell in the first row shows that 92 out of 413 Australian-accented utterances used the codebook belonging to England during decoding.

equal number of beam slots leading to eventual under-utilization of beam slots. Our proposed joint inference in the $\mathbb{B}_3$ setting is an effective compromise of $\mathbb{B}_1$ and $\mathbb{B}_2$, incurring similar inference overheads as $\mathbb{B}_2$ and achieving a performance closer to the $\mathbb{B}_1$ setting.

## 6 Discussion and Analysis

**Codebook Utilization:** Figure 2 is a heatmap showing which accent codebook is used by the joint beam-search algorithm in generating the best ASR hypothesis for the test utterances. Across seen accents, we see a diagonal dominance, indicating that seen accents show a preference for their respective codebooks. This effect is especially strong for Australia, England and Scotland accents. US and Canada, on the other hand, have examples evenly

divided among each other. Among unseen accents, the Australia-specific codebook is picked up most by New Zealand test utterances.

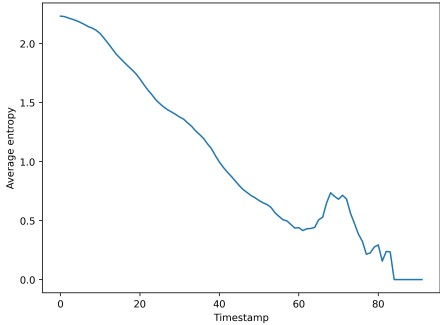

Figure 3: Progression of average test entropy of the probability distribution across seen accents.

**Active Accents during Joint Beam-search:** Using joint beam-search decoding, it is possible for samples from different accents to get pruned in early iterations and only one or two dominant accents to be active from the start. To check for this, we compute the distribution of samples in the beam across the five seen accents and plot the average entropy of this distribution across all test instances in Figure 3. It is clear that four to five seen accents are active until time-step 20, after which certain accents gain more prominence. Figure 4 shows both the probabilities across seen accents appearing in the beam for a single Wales-accented test sample, along with the entropy of this distribution. This shows how nearly all accents are active at the start of the utterance, with England becoming the dominant accent towards the end.

**Alternatives to Joint Beam-search:** We also explore two alternatives to learning accent labels within the ASR model itself: i) We jointly trained an accent classifier with ASR. During inference, this classifier provides pseudo-accent labels across seen accents that we use to choose the codebook. ii) We adopted a gating mechanism inspired by Zhang et al. (2021) that adds a learnable gate to each codebook entry. Unlike our current deterministic policy of picking a fixed subset of codebook entries, the learned gates are trained jointly with ASR to pick a designated codebook entry corresponding to the underlying accent of the utterance. During inference, the learned gates determine the codebook entries to be used for each encoder layer. Both these techniques performed better than the Conformer baseline but were equivalent in performance to the DAT approach (Das et al., 2021b).

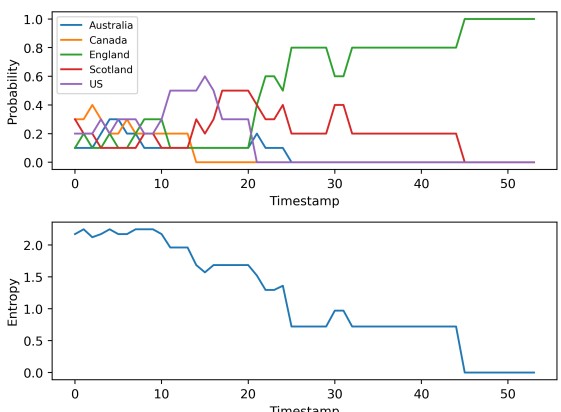

Figure 4: Progression of the probability/entropy across seen accents for a single Wales-accented test sample.

We hypothesize that this could be due to the lack of a strong accent classifier (or a lack of appropriate learning in the gates to capture accent information). Our joint beam-search decoding bypasses this requirement by searching across all seen accents.

**Why do we see Performance Improvements on Unseen Accents?** For test utterances from unseen accents, our model is designed to choose (seen) accent codebooks that best fit the underlying (unseen) accent. It is somewhat analogous to how humans use familiar accents to tackle unfamiliar ones (Anderson, 2018; Levy et al., 2019). During inference, our model searches through seen accent codebooks and chooses entries that are most like the unseen accents in the test instances.

## 7 Conclusion

In this work, we propose a new end-to-end technique for accented ASR that uses accent-specific codebooks and cross-attention to achieve significant performance improvements on seen and unseen accents at test time. We experiment with the Mozilla Common Voice corpus and show detailed ablations over our design choices. We also empirically analyze whether our codebooks encode information relevant to accents. The effective use of codebooks for accents opens up future avenues to encode non-semantic cues in speech that affect ASR performance, such as types of noise, dialects, emotion styles of speech, etc.

## Acknowledgements

The second and third authors gratefully acknowledge financial support from a SERB Core Research Grant, Department of Science and Technology, Govt of India on accented speech processing.

## 8 Limitations

We identify a few key limitations of our proposed approach:

- The codebook size is a hyperparameter that needs to be finetuned for each task.

- We currently employ accent-specific codebooks, one for each accent. This does not scale very well and also does not enable sharing of codebook entries across accent codebooks. Instead, we could use a single (large) codebook and use learnable gates to pick a subset of codebook entries corresponding to the underlying accent of the utterance.

- Our proposed joint beam-search leads to a $16\%$ increase in computation time at inference. This can be made more efficient as part of future work.

- Our joint beam-search allows for each utterance at test-time to commit to a single seen accent. However, parts of an utterance might benefit from one seen accent, while other parts of the same utterance might benefit from a different seen accent. Such a mix-and-match across seen accents is currently not part of our approach. Accommodating for such effects might improve our model further.

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

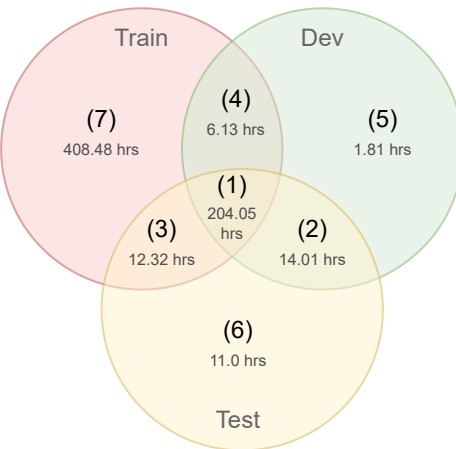

Figure 5: Illustration of the transcript-wise overlap between the `train`, `test`, and `dev` sets in terms of durations. **(1)** represents the duration of the group of utterances whose transcripts are present in all three splits. **(7)** denotes duration of examples having transcripts found only in the `train` set.

## A  Dataset Curation

To build MCV_ACCENT-600, we group the examples from MCV_ACCENT into seven buckets while preserving speaker disjointedness across the `train`, `dev`, and `test` sets.

The buckets are visualized in Figure 5; bucket (4) refers to utterances that have exactly the same transcript but different speakers appearing across the train and dev sets. We wanted to include some transcript overlap across all combinations of `train`, `test`, and `dev` splits, since the model could learn accent information from samples with the same transcripts and different underlying accents. We note that a majority of the dev and test samples are disjoint in both speakers and transcripts from the training set for a true evaluation that does not benefit from having seen the same transcripts during training.

To create such a split, we loop over all the accents, and for every seen accent $a_{seen}$, we first filter out examples with transcripts that have been previously dealt with and then split the remaining unique transcripts from $a_{seen}$ into seven buckets. For every bucket $b$, transcripts from $b$ are further divided into $n$ groups where $n$ is the number of benefactors for that bucket. As an example, for bucket (1) the value of $n$ is 3. Let $x^i$ be an utterance spoken by speaker $s^j$, which is put into the `train` set by bucket (1). Then to maintain speaker disjointedness, we put all utterances spoken by $s^j$ into the `train` set. The transcripts of these utter-

| Accent | Ground Truth | Experiment | Sentence |
|---|---|---|---|
| Australia | where is your father | Base | where is youfada |
| | | DAT | where is youfada |
| | | CA | where is your father |
| Canada | putting a pool under this floor was a great idea | Base | put it a pool into this floor was a greek of idea |
| | | DAT | put his pool under this floor was a great of itea |
| | | CA | putting a pool under this floor was a great idea |
| England | will you breakfast with me | Base | will you break for swimming |
| | | DAT | will you bright for study |
| | | CA | will you breakfast with me |
| Scotland | elsa knitted furiously | Base | elsa knitted futiously |
| | | DAT | elsa knitted fudiously |
| | | CA | elsa knitted furiously |
| US | how long since weve seen each other | Base | a long since weve seen each other |
| | | DAT | our longsons would seen each other |
| | | CA | how long since weve seen each other |
| Africa | this made them even richer | Base | this might them even richer |
| | | DAT | this might them even richard |
| | | CA | this made them even richer |
| Hongkong | he won a worldwide reputation in his special field | Base | he won a world white we potason in his special field |
| | | DAT | he won a world white repetition in his special field |
| | | CA | he won a worldwide reputation in his special field |
| India | can you play nineties music from paul kelly | Base | canuble nadis music from policy |
| | | DAT | canuple ninety is music from policy |
| | | CA | can you play nineties music from pole kelly |
| Ireland | this award is given in three different categories | Base | this award is given in the dream different categories |
| | | DAT | this award is given in the tree giffins categords |
| | | CA | this award is given in three gifting categories |
| Malaysia | it just entered my mind at that moment isaac said | Base | they just enter my mind at that moment eyes accid |
| | | DAT | it just enter my mind at that moment eyes accid |
| | | CA | it just entered my mind at that moment isac said |
| Newzealand | a content delivery network is necessary | Base | a content delivery mid luke and disincessary |
| | | DAT | a content delivery night blue and businecessary |
| | | CA | a content delivery network does necessary |
| Singapore | apple google amazon and facebook are often described as tech giants | Base | appal google amazon and face book are often described as the giants |
| | | DAT | appalo google amazon and face book are often described as tack giants |
| | | CA | apple google amazon and facebook are often described as tech giants |
| Philippines | how many layers of irony are you on | Base | how many layers of iron are you on |
| | | DAT | how many layers of ironea you are |
| | | CA | how many layers of irony are you on |
| Wales | nine rows of soldiers stood in a line | Base | minros of soldiers stood in a line |
| | | DAT | miners of soldiers stood in a line |
| | | CA | nine rows of soldiers stood in a line |

Table 11: Comparison of the predictions from the Conformer (labeled as Base), DAT, and our proposed system on a few test utterances. █ is used to highlight words that are correctly predicted, whereas █ highlights predictions that are correct but contain minor mistakes such as unwanted spaces. Similarly, █ denotes incorrectly predicted words, whereas █ is used to indicate words that are incorrect but are somewhat closer to the underlying transcript such as having similar prefixes.

ances are ignored while processing the remaining buckets and accents. For unseen accent $a_{unseen}$, all the examples with transcripts that are not yet processed are put into bucket $(6)$.

The dev and test sets built this way contain around 68 and 130 hours of data. We further randomly sample 25% and 15% from these sets, respectively. To generate MCV_ACCENT-100 split, we randomly sample 14% from the MCV_ACCENT-600 set.

## B  Additional Results

Table 12 shows word error rates (WERs) comparing all experiments done using MCV_ACCENT-100 dataset. Similarly, Table 13 shows word error rates comparing all experiments done using MCV_ACCENT-600 dataset.

## C  Comparison of Predictions

Table 11 highlights a few examples where our proposed system performs significantly better than the Conformer baseline and DAT systems.

| Method | Aggregated | | | Seen Accents | | | | | Unseen Accents | | | | | | | | |
|---|---|---|---|---|---|---|---|---|---|---|---|---|---|---|---|---|---|
| | Overall | Seen | Unseen | AUS | CAN | UK | SCT | US | AFR | HKG | IND | IRL | MAL | NWZ | PHL | SGP | WLS |
| Transformer | 22.68 | 17.34 | 28.01 | 18.11 | 17.81 | 19.73 | 18.50 | 16.31 | 25.86 | 31.97 | 35.39 | 25.25 | 36.25 | 23.83 | 31.50 | 38.78 | 21.04 |
| Conformer (Base) | 18.87 | 14.05 | 23.67 | 13.82 | **15.02** | 15.74 | **13.36** | 13.30 | 21.47 | 27.18 | 29.39 | 21.38 | 32.20 | 19.86 | 26.13 | 34.69 | 17.88 |
| I-vector sum | 18.87 | 14.15 | 23.58 | 13.88 | 15.02 | 16.07 | 14.62 | 13.31 | 21.66 | 27.18 | 29.53 | 21.18 | **31.72** | 19.33 | 27.22 | **33.81** | 17.98 |
| Base + Classifier | 18.91 | 14.12 | 23.69 | 14.73 | 15.10 | 16.08 | 13.72 | 13.19 | 21.83 | 28.13 | **29.15** | 21.46 | 32.97 | 19.38 | 26.51 | 34.24 | 18.08 |
| DAT | **18.70** | **14.00** | **23.38** | **13.30** | 15.30 | **15.72** | 15.52 | **13.15** | **21.15** | **26.95** | 29.53 | **21.15** | 32.16 | **19.22** | **26.03** | 34.43 | **17.93** |
| $CA_{L\in(1,...,12)}(P=25)$ | 18.33 | 13.76 | 22.89 | 13.05 | 14.90 | 15.33 | 10.92 | 13.12 | 20.82 | 26.41 | 29.29 | 20.70 | 31.55 | 18.56 | 26.10 | 33.30 | 16.58 |
| $CA_{L\in(1,...,12)}(P=50)$ | 18.22 | 13.57 | 22.86 | 11.54 | 14.81 | 14.91 | 9.66 | 13.15 | 20.95 | 25.66 | 29.15 | 20.72 | 30.87 | 18.47 | 25.81 | 33.68 | 17.92 |
| $CA_{L\in(1,...,12)}(P=100)$ | 18.36 | 13.85 | 22.86 | 12.89 | 14.91 | 15.46 | 10.92 | 13.24 | 20.77 | 25.89 | 28.87 | 20.41 | 32.44 | 18.76 | 26.24 | 32.93 | 17.77 |
| $CA_{L\in(1,...,12)}(P=200)$ | 18.41 | 13.69 | 23.12 | 13.00 | 14.81 | 15.06 | 11.10 | 13.12 | 21.57 | 26.78 | 28.30 | 20.93 | 30.95 | 18.97 | 25.97 | 33.17 | 17.88 |
| $CA_{L\in(1,...,12)}(P=500)$ | 18.39 | 13.68 | 23.09 | 12.04 | 14.93 | 15.40 | 11.10 | 13.05 | 21.22 | 26.52 | 28.77 | 20.57 | 33.13 | 18.78 | 25.97 | 33.81 | 18.39 |
| $CA_{L\in(1,...,4)}(P=50)$ | 18.30 | 13.95 | 22.64 | 13.22 | 15.53 | 15.49 | 10.74 | 13.24 | 20.79 | 26.06 | 28.58 | 20.52 | 31.43 | 17.87 | 26.27 | 32.98 | 17.72 |
| $CA_{L\in(1,...,8)}(P=50)$ | 18.31 | 13.86 | 22.75 | 13.14 | 15.07 | 15.76 | 10.56 | 13.12 | 20.50 | 25.52 | 28.70 | 21.03 | 30.70 | 18.53 | 25.59 | 33.10 | 18.03 |
| $CA_{L\in(9,...,12)}(P=50)$ | 18.92 | 14.24 | 23.59 | 13.30 | 15.36 | 15.53 | 13.27 | 13.67 | 21.50 | 27.18 | 28.89 | 21.61 | 32.52 | 18.98 | 26.64 | 34.43 | 19.59 |
| $CA_{L\in(5,...,12)}(P=50)$ | 18.45 | 13.84 | 23.05 | 12.37 | 15.43 | 15.42 | 10.92 | 13.18 | 21.08 | 26.75 | 28.40 | 20.60 | 31.76 | 18.68 | 26.52 | 34.00 | 18.13 |
| $CA_{L\in(1,...,12)}(P_{rand}=50)$ | 18.30 | 13.65 | 22.95 | 12.34 | 15.19 | 14.89 | 11.64 | 13.06 | 20.77 | 25.57 | 29.48 | 20.98 | 31.88 | 18.62 | 25.87 | 33.28 | 17.82 |

Table 12: Comparison of the performance(WER % ) of all the experiments mentioned for MCV_ACCENT-100. Numbers in bold denote the best across baselines, and the green highlighting ▨ denotes the best across all the experiments. Ties are broken using overall WER. $CA_{L\in(i,...,j)}(P=k)$: Codebook attend - Cross Attention applied at all layers from $i$ to $j$ with $k$ entries per accent codebook. $CA_{L\in(i,...,j)}(P_{rand}=k)$: Similar to the previous setup, but with codebooks frozen during training.

| Method | Aggregated | | | Seen Accents | | | | | Unseen Accents | | | | | | | | |
|---|---|---|---|---|---|---|---|---|---|---|---|---|---|---|---|---|---|
| | Overall | Seen | Unseen | AUS | CAN | UK | SCT | US | AFR | HKG | IND | IRL | MAL | NWZ | PHL | SGP | WLS |
| Conformer (Base) | 9.75 | **6.04** | 13.46 | 4.95 | **6.81** | 7.15 | 4.42 | **5.64** | 11.81 | 16.50 | 14.90 | **12.61** | 20.43 | 10.49 | **15.74** | 21.26 | **7.98** |
| I-vector | 10.05 | 6.40 | 13.69 | 4.67 | 7.53 | 7.43 | 4.06 | 6.04 | 12.05 | 17.45 | 14.76 | 13.08 | 20.31 | 10.57 | 15.82 | **21.22** | 8.91 |
| MTL | 10.02 | 6.33 | 13.70 | 5.30 | 7.59 | 7.39 | **3.97** | 5.86 | 12.19 | 16.30 | **14.25** | 13.23 | 19.58 | 10.64 | 16.30 | 21.54 | 8.50 |
| DAT | **9.73** | 6.12 | **13.33** | **4.56** | 7.50 | **6.87** | 4.96 | 5.74 | **11.76** | 16.19 | 14.62 | 12.96 | **18.97** | 9.91 | 15.84 | 21.50 | 8.13 |
| $CA_{L\in(1,...,12)}(P=50)$ | 9.63 | 6.22 | 13.03 | 4.67 | **7.36** | 7.11 | 3.07 | **5.90** | 11.63 | 15.64 | 14.06 | 12.48 | 18.97 | **9.73** | 15.60 | 21.05 | 8.29 |
| $CA_{L\in(1,...,12)}(P=200)$ | 9.59 | 6.20 | 12.98 | 4.92 | 7.47 | 6.83 | 3.52 | 5.91 | **11.47** | 15.96 | **13.87** | 12.25 | 19.30 | 9.98 | **15.17** | 20.90 | **8.08** |
| $CA_{L\in(1,...,12)}(P=500)$ | **9.55** | **6.19** | **12.92** | **4.40** | 7.60 | **6.67** | **2.62** | 5.99 | 11.57 | **15.18** | 14.13 | **12.20** | **17.84** | 10.00 | 15.34 | **20.71** | 8.39 |

Table 13: Comparison of the performance(WER % ) of all the experiments mentioned for MCV_ACCENT-600. We follow the same notation as in Table 12.