# OpenReview forum: "Accented Speech Recognition With Accent-specific Codebooks"
_EMNLP/2023/Conference — EMNLP 2023 Main_

### Official Review · Reviewer_78aE · 2023-07-30

**Typos Grammar Style And Presentation Improvements:** Line 307, Figure 2 is wrongly referen…
**Soundness:** 4

**Excitement:**

4: Strong: This paper deepens the understanding of some phenomenon or lowers the barriers to an existing research direction.

**Missing References:**

There is a rich body of work done during the traditional ASR times on accented speech. Would be nice if the authors cite those as well. Here are some such papers:

- "German regional variants - a problem for automatic speech recognition?" by N. Beringer (1998)
- "Towards acoustic model unification across dialects" by M. Elfeky (2016)
- "Multi-accent speech recognition with hierarchical grapheme based models" by K. Rao (2017)

The authors also missed an influential accented E2E paper:

- "Multi-dialect speech recognition with a single sequence-to-sequence model" by B. Li (2018). A similar system would have also been a strong baseline for the system proposed, I feel.

**Paper Topic And Main Contributions:**

Existing ASR systems are vulnerable to accent variation, especially so for those accents that are underrepresented in training datasets (e.g., African accented English). This work proposed a system that uses learnable accent codebooks + joint beam search (searches over all accents simultaneously) to solve this issue. The proposed solution outperforms strong baselines (upto 37% in relative WER) on a standard benchmark, i.e, common voice.

**Questions For The Authors:**

- [A] How come this method is doing so well for unseen accents? More discussion on this in the paper would be nice to see.
- [B] Hong Kong, Indian accents should prefer England accent due to historical (and also geographical reasons to some extent) reasons. However, they prefer US. What could be the reason for this?
- [C] Randomly initialised non-learnable code-books experiments is intriguing. This makes me wonder whether the codebooks are used in the way that we think the model is using or if there is more to it. Would like the authors to comment more about why such a system works.
- [D] Could the authors add I-vector based system and MTL system results in Table 5 (on 600 hour common voice) and also in Table 4 (for zero-shot setting)? I suspect the MTL/I-vector model to do well when trained on 600 hour dataset. One of them could be the best performing model in this setting.

**Reasons To Accept:**

- Clear motivation and a well written paper.
- Really enjoyed the way the authors grouped the works into accent aware and accent-agnostic systems, thereby covering a wide-range of papers!
- The proposed system is interesting -- codebooks has not been applied to accented E2E ASR and is very suitable for the task. Joint beam search proposed by authors is a neat trick!
-  The work was systematically executed with experiments with a wide-range of hyperparameters and accents.
- The system does exceedingly well in a zero-shot setting!

**Reasons To Reject:**

- Improvements due to this architecture start to go down as the data increases even to the point where other methods (just Conformer baseline!) start doing well on seen accents.
- It is unclear how the model would perform when the dataset is accent-balanced.
- The codebooks size may explode if we (eventually) want to model 100s of accents (for example, consider Africa -- people of Africa alone speak in > 200 accents), if we dedicate one codebook per accent.

**Reproducibility:**

4: Could mostly reproduce the results, but there may be some variation because of sample variance or minor variations in their interpretation of the protocol or method.

**Reviewer Confidence:**

4: Quite sure. I tried to check the important points carefully. It's unlikely, though conceivable, that I missed something that should affect my ratings.

---

> ### Author Rebuttal · Authors · 2023-08-29
>
> Thank you for your valuable time and effort in reviewing our paper and providing us with constructive feedback. Below, we address all the questions you raised in your review.
>
> ---
>
> > **How come this method is doing so well for unseen accents? More discussion on this in the paper would be nice to see.**
>
> For test utterances with unseen accents, our model is designed to choose (seen) accent codebooks that best fit the underlying (unseen) accent. It is somewhat analogous to how humans use familiar accents to tackle unfamiliar ones (supported by Section 5.4 of the book "[Essentials of Linguistics](https://ecampusontario.pressbooks.pub/essentialsoflinguistics/chapter/5-3-attitudes-about-accents/)" by Catherine Anderson and in the paper "[Processing of unfamiliar accents in monolingual and bilingual children](https://www.cambridge.org/core/journals/journal-of-child-language/article/processing-of-unfamiliar-accents-in-monolingual-and-bilingual-children-effects-of-type-and-amount-of-accent-experience/7B22B1E7C59E024EEBC4025A6E424B29)" published in the [Journal of Child Language](https://www.cambridge.org/core/journals/journal-of-child-language) by Helena et al). During inference, our model searches through seen accent codebooks and chooses entries that are most like the unseen accents in the test instances.
>
> ---
>
> > **Hong Kong, and Indian accents should prefer England accents due to historical (and also geographical reasons to some extent) reasons. However, they prefer US. What could be the reason for this?**
>
> From listening to a sample of US accented utterances in the MCV dataset, we find them to be very heterogeneous including both native speakers of English and speakers of English originally from India, China, etc. with native fluency but traces of non-native accents appearing in their speech. This could be one reason for the Indian, Hong Kong accented samples preferring US codebooks.
>
> You can listen to a few examples from this [anonymized website](https://emnlp2023.notion.site/emnlp2023/Presence-of-heterogenity-in-the-US-split-of-the-Mozilla-Commonvoice-Dataset-1-9147652c526d4baa8c153f6927d9f384).
>
> ---
>
> > **Randomly initialized non-learnable code-books experiments are intriguing. This makes me wonder whether the codebooks are used in the way that we think the model is using or if there is more to it. Would like the authors to comment more about why such a system works.**
>
> Codebooks could be made learnable to encode additional information about the underlying accents, or the codebooks could be randomly initialized and frozen during training. The model will learn to adapt to the frozen codebook entries and pick suitable entries specific to different accents. A similar scheme involving randomly initialized frozen codebook entries was previously found to be beneficial for ASR in the BEST-RQ self-supervised algorithm for speech recognition proposed by Chiu et al., “Self-Supervised Learning with Random-Projection Quantizer for Speech Recognition”, ICML 2022.
>
> ---
>
> > **Could the authors add I-vector based system and MTL system results in Table 5 (on 600 hour common voice) and also in Table 4 (for zero-shot setting)? I suspect the MTL/I-vector model to do well when trained on 600-hour dataset. One of them could be the best-performing model in this setting.**
>
> Below we report results using the i-vector and MTL systems in the zero shot setting (Table 4):
>
> | Model Name | Overall | Arabic | Hindi | Korean | Mandarin | Spanish | Vietnamese |
> | :--- | :---: | :---: | :---: | :---: | :---: | :---: | :---: |
> | Conformer | 33.3 | 30.4 | 30.4 | 26.9 | 37.9 | 30.3 | 43.5 |
> | DAT | 33.5 | 30.7 | 30.8 | 26.8 | 38.3 | 30.1 | 43.9 |
> | i-vector | 33.6 | 31.0 | 31.2 | 27.2 | 38.0 | 30.4 | 43.9 |
> | MTL | 33.4 | 30.4 | 30.6 | 26.9 | 38.7 | 30.1 | 43.7 |
> | CA | **32.6** | **29.5** | **30.4** | **26.2** | **37.1** | **29.3** | **42.8** |
>
> We observe that both i-vector and MTL underperform compared to our model CA.
>
> The i-vector and MTL-based systems for the 600-hour dataset are still running and could not be completed within the rebuttal window. We will share these results during the discussion phase. We would additionally like to present more numbers with our model in the 600-hour setting using larger numbers of codebooks that leads to further reductions in WER.
>
> | Model Name | Overall WER | Seen WER | Unseen WER |
> | :--- | :---: | :---: | :---: |
> | Baseline | 9.75 | **6.04** | 13.46 |
> | DAT | 9.73 | 6.12 | 13.33 |
> | $\text{CA}_{L \in (1,2,\ldots 12)} (P=50)$ | 9.63 | 6.22 | 13.03 |
> | $\text{CA}_{L \in (1,2,\ldots 12)} (P=200)$ | 9.59 | 6.20 | 12.98 |
> | $\text{CA}_{L \in (1,2,\ldots 12)} (P=500)$ | **9.55** | 6.19 | **12.92** |
>
> > **How does the model perform when the dataset is accent-balanced?**
>
> To check the effectiveness of our approach on a balanced dataset, we created a 100-hour accent balanced data split. The results are as follows:
>
> | Model Name | Overall WER | Seen WER | Unseen WER |
> | :--- | :---: | :---: | :---: |
> | Baseline | 19.30 | 14.73 | 23.86 |
> | Ours | **18.88** | **14.61** | **23.13** |
>
>
> Even on the balanced dataset, our architecture shows a **statistically significant improvement** (at $p = 0.005$) compared to the baseline.
>
> ---
>
> > **The codebooks size may explode if we (eventually) want to model 100s of accents**
>
> Indeed, this is a limitation with the current proposal that we plan to address in future work. One direction we are currently exploring is introducing codebooks with shared/tied entries (analogous to tied-HMMs). Enabling such codebook tying will substantially reduce the number of codebook entries and avoid codebook explosion.

---

### Official Review · Reviewer_ndKt · 2023-08-04

**Soundness:** 3

**Excitement:**

4: Strong: This paper deepens the understanding of some phenomenon or lowers the barriers to an existing research direction.

**Paper Topic And Main Contributions:**

The paper introduces a method to improve speech recognition accuracy for various accents by introducing accent-specific codebooks which are trained based on known accents. The codebook information is provided to the ASR encoder via cross-attention. During recognition, the authors propose a joint beam search algorithm which iterates over all possible accents. The experimental results indicate small, but statistically significant improvements for seen and unseen accents. Several ablation studies augment the paper.

**Reasons To Accept:**

The paper is well written, the proposed approach is new for accent-aware ASR and the experiments are comprehensive.

**Reasons To Reject:**

none

**Reproducibility:**

3: Could reproduce the results with some difficulty. The settings of parameters are underspecified or subjectively determined; the training/evaluation data are not widely available.

**Reviewer Confidence:**

3: Pretty sure, but there's a chance I missed something. Although I have a good feel for this area in general, I did not carefully check the paper's details, e.g., the math, experimental design, or novelty.

---

> ### Author Rebuttal · Authors · 2023-08-29
>
> Thank you for your positive feedback on our paper. We appreciate your valuable time and effort in reviewing our work. We are happy to provide any additional experiments or clarifications, which might further strengthen our submission.

---

### Official Review · Reviewer_tMje · 2023-08-06

**Soundness:** 3

**Excitement:**

4: Strong: This paper deepens the understanding of some phenomenon or lowers the barriers to an existing research direction.

**Paper Topic And Main Contributions:**

- A technique to improve the robustness of ASR for accented speech, with  learnable accented codebooks using cross-attention mechanism.
- A modified beam search algorithm, which allocate  different beam widths to different seen accents, based on the hypothesis scores.
- The potential release of train/dev/test splits for multi-accents ASR from common voice corpus.

**Questions For The Authors:**

1. Could you give more details on the model architectures, size, computation for those reported in Table 3, 4, 5?
2. The codebook construction is quite simple, the fixed number of code entries may not be optimal as the accent data is imbalance. Could you comment on this?
3. Could you give some statistics on the number distinct accents through several time steps of the modified beam search?

**Reasons To Accept:**

- Based on the experimental results, the proposed approach is quite effective for both seen and unseen accents.
- The paper is generally well-written, with comprehensive ablation studies. The results from Table 7 is interesting, as it potentially highlights the correlation between accents and acoustic features.

**Reasons To Reject:**

- The performance reported in Table 3,4 and 5 may not be entirely fair, as the model size, computation complexity and memory usage are not taken into consideration. It could be argued that the better performance is due to larger model size, more computational intensive, etc.
- The modified beam search algorithm (Algorithm 1) is not guaranteed to search through all the seen accent codebooks, several initial frames with high scores may lead to premature pruning of all low-score accents. And this would behave like using a classifier to select the accent codebook.
- In Figure 2, for seen accents, the off-diagonal values are quite close to the diagonal ones, which may hint that there are strong correlations among the learnt accented codebooks.

**Reproducibility:**

4: Could mostly reproduce the results, but there may be some variation because of sample variance or minor variations in their interpretation of the protocol or method.

**Reviewer Confidence:**

4: Quite sure. I tried to check the important points carefully. It's unlikely, though conceivable, that I missed something that should affect my ratings.

---

> ### Author Rebuttal · Authors · 2023-08-29
>
> Thank you for your valuable time and effort in reviewing our paper and providing us with useful insights. Below, we address all the questions and concerns raised in your review.
>
> ---
>
> > **Details on the model architectures, size, and computation for those reported in Table 3, 4, 5.**
>
> To eliminate the possibility that the improvement in our proposed model is solely due to an increased number of parameters, we conduct baseline experiments in two different ways to make it commensurate with the number of parameters in our proposed model.
>
> `Please note: We use the same model specifications for all the experiments mentioned in Tables 3, 4, and 5. Below, we report numbers only for the setting used in Table 3.`
>
> 1. **Baseline**: Baseline conformer model reported in Table 3 (referred to as *Conf.*).
> 2. **M1**: Baseline conformer model with increased encoder units (from $2048 \rightarrow 2320$).
> 3. **M2**: Baseline conformer model with increased attention dimension (from $256 \rightarrow 272$).
> 4. **Ours**: Our proposed cross-attention model reported in Table 3 (referred to as *CA*).
>
> | Model Name | Number of Parameters | Overall WER | Seen WER | Unseen WER |
> | :--- | :---: | :---: | :---: | :---: |
> | Baseline | 43M | 18.87 | 14.05 | 23.67 |
> | M1 | 46M | 18.89 | 14.02 | 23.74 |
> | M2 | 46M | 18.77 | 14.02 | 23.51 |
> | Ours |46M | **18.22** | **13.57** | **22.86** |
>
> Even with an increase in parameters for the baseline models, we still see a **statistically significant improvement** (at $p<0.001$) with our system compared to M2.
>
> ---
>
> > **Fixed number of code entries may not be optimal as the accent data is imbalanced.**
>
> To check the effectiveness of our approach on a balanced dataset, we created a 100-hour accent balanced data split. The results are as follows:
>
> | Model Name | Overall WER | Seen WER | Unseen WER |
> | :--- | :---: | :---: | :---: |
> | Baseline | 19.30 | 14.73 | 23.86 |
> | Ours | **18.88** | **14.61** | **23.13** |
>
> Even on the balanced dataset, our architecture shows a **statistically significant improvement** (at $p = 0.005$) in comparison to baseline.
>
> In this work, we use a fixed number of codebooks yielding significant WER reductions. For accent-imbalanced datasets, using a variable number of codebooks per accent might indeed achieve better results. This is a non-trivial modification to the codebook architecture that we leave for future work.
>
> ---
>
> > **Statistics on the number of distinct accents through several time steps of the modified beam search.**
>
> Please refer to [these plots](https://emnlp2023.notion.site/Plots-5db415a2e2f14776b2bb10f064f69a84?pvs=4). Across decoding time-steps, we show 1) the entropy of the probability distribution across seen accents averaged across all test samples and 2) the probability of a seen accented sample appearing in the beam for a single test example. From the entropy plot, it is clear that at least four or all five seen accents are active until time-step 20 after which certain accents gain more prominence. The probability plot shows how (nearly) all accents are in the running until time-step 20, after which it slowly converges to one or two accents.
>
> ---
>
> > **In Figure 2, for seen accents, the off-diagonal values are quite close to the diagonal ones, hinting at strong correlations among the learnt accented codebooks.**
>
> The perceived correlation among the learned codebooks can be attributed to high correlation between certain seen accents. Primarily, as mentioned in our analysis in Section 6, the US and Canada accents in MCV are perceptually very similar, resulting in our model picking Canada's codebooks for US examples and vice-versa. This shared preference between US and Canada leads to certain off-diagonal seen accents’ codebooks appearing to be close to the diagonal one (e.g., Scotland for Canada). This perceived correlation between off-diagonal and diagonal entries would not appear if the choice of codebooks across US and Canada accented utterances skewed towards either US or Canada, respectively.

---

### Meta-Review · Area_Chair_DnFj · 2023-09-08

**Recommendation:** 4

**Metareview:**

I concur with the other reviewers and believe this paper's idea is valuable; I recommend acceptance.

---

### Decision · Program_Chairs · 2023-10-07

**Decision:**

Accept-Main

**Comment:**

I concur with the other reviewers and believe this paper's idea is valuable; I recommend acceptance.